# Nutritional Value and Contaminant Risk Assessment of Some Commercially Important Fishes and Crawfish of Lake Trasimeno, Italy

**DOI:** 10.3390/ijerph17072545

**Published:** 2020-04-08

**Authors:** Raffaella Branciari, Raffaella Franceschini, Rossana Roila, Andrea Valiani, Ivan Pecorelli, Arianna Piersanti, Naceur Haouet, Marisa Framboas, David Ranucci

**Affiliations:** 1Department of Veterinary Medicine, University of Perugia, Via San Costanzo 4, 06126 Perugia, Italy; david.ranucci@unipg.it; 2Department of Sustainability Engineering, Guglielmo Marconi University, 00193 Rome, Italy; raffaella.franceschini@gmail.com; 3Istituto Zooprofilattico Sperimentale dell’Umbria e delle Marche “Togo Rosati”, Via G. Salvemini 1, 06126 Perugia, Italy; a.valiani@izsum.it (A.V.); i.pecorelli@izsum.it (I.P.); a.piersanti@izsum.it (A.P.); mn.haouet@izsum.it (N.H.); m.framboas@izsum.it (M.F.)

**Keywords:** freshwater fish and crawfish, fatty acids and cholesterol, metals, PCBs, pesticides, dietary intake, dietary exposure, risk benefit

## Abstract

The aim of our study was to describe the balance between health benefits and risks associated with the consumption of crawfish and nine fish species from lake Trasimeno. We thus determined both fatty acid profiles (particularly, eicosapentaenoic (EPA) and docosahexaenoic (DHA) acids) and chemical pollutants (some polychlorinated biphenyls, pesticides, and heavy metals) in fish muscle tissues. The contents of all fatty acids varied significantly among species. Sand smelt, carp, and tench, which have a high fat content, contained considerable amounts of EPA and DHA; lean fish, like perch, pike, and largemouth bass, which have relatively high percentages of the predominant n-3 fatty acids EPA and DHA, showed lower amounts of these fatty acids because of their low lipid contents. Some species contributed strongly to the Dietary Reference Intake (RDI) of EPA and DHA. The contribution of lean fish to the RDI of EPA and DHA was more limited. The concentrations of all contaminants in fish muscle tissues were lower than the regulatory limits, demonstrating the safety of the environmental conditions of the lake. The contribution to health-based reference values and benefit–risk quotients indicated that the health benefits of consumption of fish from lake Trasimeno outweigh the potential risks.

## 1. Introduction

The nutritional benefits of fish consumption are attributable to the high-value proteins and unsaturated fatty acids, as well as certain minerals and vitamins, contained in fish. Particularly, fish are considered a very important source of n-3 polyunsaturated fatty acids (PUFA), especially eicosapentaenoic (EPA, C 20:5n-3) and docosahexaenoic (DHA, C 22:6n-3) [1,2]. Several clinical studies have demonstrated the relationship between n-3 PUFA intake and both the maintenance of physiological functions (optimal cardiovascular system, brain and vision functioning) and the prevention of certain health conditions (coronary artery disease and cancer), as well as the positive effects of n-3 PUFA consumption on some diseases (arthritis, hypertension, and diabetes mellitus) [3,4]. It is generally recognized that freshwater fish meat generally contains lower proportions of C 20, C 22, and n-3 PUFA but higher amounts of C 18 and n-6 PUFA, in particular 18:2n-6, than marine fish [5]. These differences are mainly attributable to the diverse fatty acid composition of organisms constituting the marine food web compared with those of the freshwater food web [1,5,6].

Nevertheless, Li et al. [2] found that some freshwater fishes, as do marine fish, possess a higher content of n-3 PUFA, EPA, and DHA in the edible meat and related this to the food habits and trophic levels of the fishes. Despite the widely acknowledged health benefits of fish consumption, some concerns have been raised in relation to the presence of undesired chemical pollutants in fish meat, such as heavy metals, methyl mercury (MeHg), polychlorinated biphenyls (PCBs), dioxins and dioxin-like contaminants (DLCs), organochlorine pesticides, and other environmental contaminants [7,8,9]. Fish products, indeed, are particularly susceptible to contamination, especially those from lentic ecosystems characterized by slow water exchange and elevated anthropization [10]. Although persistent organic pollutant production and use were restricted following the Stockholm Convention in 2001, the presence of pesticides, PCBs, and heavy metals in aquatic ecosystems represents one of the most debated environmental issues, due to the contaminants’ ubiquitous presence in water and soil, their accumulation along the food web, and their potential threat to public health [11,12]. The European Food Safety Authority, due to the variety of fish species consumed across Europe, recommends that each country consider its own pattern of fish consumption, especially the species of fish most consumed, and carefully assess the risk of exceeding the safety levels of contaminants, while also evaluating the health benefits of fish consumption [13].

In the present work, in order to define the nutritional quality and safety of nine economically important species of freshwater fish and crawfish caught in lake Trasimeno (Umbria, Central Italy), the fatty acid composition, as well as the concentrations of certain chemical pollutants (pesticides, PCBs, and metals), were defined. Furthermore, aiming to describe the balance between health benefits and risks due to fish and crawfish consumption, the intake of n-3 PUFA and the estimated daily intake (EDI) of the mentioned contaminants were assessed.

## 2. Materials and Methods

### 2.1. Aquatic Environment

Lake Trasimeno (Umbria, Italy; 43°9′11′’ N 12°15′ E) is the largest laminar lake in Italy and, more generally, is the fourth among the largest Italian lakes; furthermore, in spite of its large extension (about 128 km^2^), it is considered quite shallow (average depth: 4.72 m; max. depth: 6.3 m). This Umbrian lake is characterized by a remarkable naturalistic importance, as it is, indeed, a Site of Community Interest, Special Protection Zone and Regional Park (Figure 1).

Since the 20th century, the composition of the lake Trasimeno fauna has changed considerably due to several factors, such as eutrophication, pollution, significant hydrologic fluctuations, and the introduction of allochthonous species. Despite the above-mentioned factors, the Umbrian lake is considered a high-value ecosystem due to its wide biodiversity: its fish fauna, in fact, comprises 19 species dominated by those belonging to the Cyprinidae Family [14]. The representative phytoplankton community is characterized by seasonal fluctuations and is represented in winter by *Hyaloraphidium contortum* and *Scenedesmus ecornis,* belonging to the Cloroficeee Family, and *Leptolyngbya* spp., a member of the Cyanoficee Family, whereas in summer, the phytoplankton is almost exclusively represented by *Cylindrospermopsis raciborskii* (Cyanoficee) [15]. Similarly, the composition of zooplankton differs depending on the season: the winter population is mainly represented by numerous species of freshwater copepods belonging to the genus *Cyclops*, whereas in summer, *Daphnia galeata,* a small species of crustaceans belonging to the Order Cladocera, is the main zooplankton representative [16].

Nowadays, small-scale traditional fishing is still one of the main commercial activities of the lacustrine area, promoted by both governmental guidelines encouraging the production and consumption of local food and consumer preference for Km 0 foodstuff. Small-scale traditional fishing, as well as the consumption of fish in the area of lake Trasimeno, can be considered sustainable. In fact, the freshwater fish industry of this area is expanding into the market by exploiting the great potential of abundant undervalued species consumed as ingredients in processed food and by the adoption of long-term preservation through a cold chain system to compensate the seasonality of fish [17]. In this context, the most common edible fishes and crawfish present in lake Trasimeno were analyzed for both nutritional and safety aspects: carp (*Cyprinus carpio*, L.), golden fish (*Carassius auratus*, L.), tench (*Tinca tinca*, L.), pike (*Esox lucius*, L.), eel (*Anguilla anguilla*, L.), sand smelt (*Atherina boyer*i, R.), perch (*Perca fluviatilis*, L.), black bullhead (*Ictalurus melas*, R.), largemouth bass (*Micropterus salmoides*, L.), and crawfish (*Procambarus clarkii,* G.).

### 2.2. Chemical Composition of Fish

Fish and crawfish collected for the present study were caught from lake Trasimeno. Fishes and crawfish sampling was conducted in three different months: December 2018 and January and February 2019. After collection, all fishes were transported the same day under refrigeration to the Department of Veterinary Medicine, University of Perugia. Upon arrival, each individual was beheaded and dressed (for fish) or decarapaced (for crawfish), and the muscle tissue (edible muscle) was filleted and stored at −80 °C until analysis. Sand smelt, due to their small size (adult maximum length around 7 cm), were left whole. Four individuals of each freshwater fish species and six individuals for crawfish and sand smelt, each month, were used for the analysis.

The chemical composition and fatty acid profiles of each individual were determined according to the procedure of Branciari et al. [17,18].

### 2.3. Analytical Contaminant Determination

Samples for contaminant determination were collected by local official competent authorities from fishermen and sent to the official local laboratory located at Istituto Zooprofilattico Sperimentale dell’Umbria e delle Marche. The number of samples collected was defined according to the integrated regional control plan, and sampling was carried out randomly. Fish muscles were analyzed to assess PCB levels (as the sum of the six indicator congeners: PCB-28, -52, -101, -153, -138, and -180). A sample amount yielding roughly 1 g of fat was weighted, spiked with 2.5 ng of the six 13C12-labelled indicator PCBs, and freeze-dried (6 h). The sample powder obtained was mixed with diatomaceous earth and extracted with ASE 200 (Dionex Corporation, Sunnyvale, CA), using a mixture of hexane/acetone (1:1 v/v). The extracts, after solvent evaporation, were purified on an H_2_SO_4_-acidified Extrelut NT-3 column connected on top to an SPE silica column (1 g/6 mL), and the analytes were eluted with 13 mL of n-hexane. The purified extract was re-dissolved in 0.25 mL of a 13C12-PCB-155 iso-octane solution at 10 ng/mL and submitted to GC–MS/MS analysis (Agilent 7890A GC coupled to Agilent 7000 QqQ; Agilent Technologies, Palo Alto, CA, USA). The injection was carried out in pulsed splitless mode, and the chromatographic separation was achieved in temperature-programmed mode (120 °C, ramp to 200 °C at 20 °C/min, ramp to 270 °C at 3 °C/min, ramp to 300 °C at 15 °C/min, hold 4.67 min, total run time 34 min) on an SGE-HT8 PCB capillary column (60 mm × 0.25 mm × 0.25 μm; SGE analytical science, Ringwood Victoria, Australia), using helium at 1 mL/min as the carrier gas. The transfer line was held at 280 °C, the source at 230 °C, and the quadrupoles at 150 °C. The method was submitted to multi-level validation in intra-laboratory reproducibility conditions, following the prescription of Commission Regulation (EU) 2017/644. The procedure enabled to measure 0.10 ng/g fresh weight for each of the six PCB congeners (limit of quantification = LOQ); moreover, the analyses were conducted in isotopic dilution for all the six PCBs, and acceptable internal standards recoveries were between 60% and 120%, as requested by Regulation (EU) 2017/644 [19].

Pb, Cd, Hg, Ni, Cr, and As were analyzed in 1 g of sample after microwave digestion with 6 mL HNO3 (67%–69%, v/v), 2 mL of H2O2 (30%, v/v), and 100 µL HF (40%, v/v) (Milestone-Ethos1-HPR1000). The appropriately diluted solutions were analyzed by inductively coupled plasma mass spectrometry (ICP-MS, Elan DRCII Perkin Elmer) in standard mode using specific mass-to-charge ratios (m/z) for each element (206 + 207 + 208 Pb, 111 Cd, 202 Hg, 60 Ni, 52 Cr, 75 As). We used 103Rh as the internal standard, and quantification was matrix-matched. The analytical methods were fully validated in intra-laboratory reproducibility conditions, following Commission Regulation (EC) No 333/2007 [20]. The LOQs (mg/kg) of the method were: Pb = 0.015, Cd = 0.005, Hg = 0.025, Ni, Cr, As = 0.040. Batch-to-batch precision and accuracy were evaluated by analyzing a certified reference material (Dorm4, NIST Canada).

Pesticide analysis was conducted by the QuEChERS method, as reported. In brief, extraction was performed on homogenized samples (5.00 ± 0.05 g) with 10 mL acetonitrile (ACN). δ-lindane, triphenilphoshate, and PCB 198 were added as internal standards (IS) for organochlorine, organophosphorus (P), and pyrethroids (Py) pesticides, respectively. The extract was shaken on an orbital shaker for 20 min, and afterwards a salt mix (MgSO_4_, 4 g; NaCl, 1 g; tri-sodium citrate dehydrate, 1 g; sodium hydrogen citrate sesquihydrate, 0.5 g) was added, and the tubes were vortexed for 1 min and centrifuged for 5 min at 4000 rpm. Eight milliliters of ACN were then transferred to a clean tube and stored in a freezer overnight. Six milliliters of cold ACN were purified by d-SPE (150 mg PSA + 900 mg Mg_2_SO_4_) and centrifuged for 8 min at 3500 rpm. Finally, 1 mL of purified extract was dried in a rotary evaporator (80 °C, 150 mbar) and resuspended in 0.2 mL of isooctane. The sample solution was injected in splitless mode, and gas chromatography was carried out using an Agilent EI GC-MS/MS system equipped with a 7890A GC system, 7000B GC/MS triple QUAD, and a 7963 autosampler. Chromatographic separation was carried out through a Rtx-Pesticides2 column in temperature-programmed mode. The method was validated according to the European Commission Guidance Document SANTE\11945\2015 on Method Validation and Quality Control Procedures for Pesticide Residue Analysis in Food and Feed. Performance parameters of the method in terms of repeatability, reproducibility under intra-laboratory conditions, linearity, recovery, specificity, and sensitivity were established by validation in a single laboratory. Within-laboratory reproducibility (RSD_WLR_) combined with data from EURL-AO proficiency tests were used for the estimation of measurement uncertainty according to Document SANTE\11945\2015. The LOQ was fixed at 0.002 mg/kg for all organochlorine pesticides and dinitroaniline, and at 0.005 mg/kg for organophosphorus and pyrethroids.

### 2.4. Dietary Exposure Assessment and Risk Characterization

The EDI of metals, pesticides, and PCBs from each fish group was calculated by multiplying the respective concentration in food by the weight of fish consumed daily by an average adult weighting 70 kg, as reported elsewhere [21]. Data on freshwater fish and crawfish consumption were extrapolated from a questionnaire administered to 100 residents around lake Trasimeno. The participants were 52 females and 48 males, with ages ranging from 19 to 65 years. The questionnaire was designed to obtain information about the frequency of consumption of different products, and the responses were combined with the food portion size data reported by the Italian dietary surveys [22]. The assessors provided their consent prior to the tests, they did not receive any incentives for their participation, and the questionnaires were returned anonymously. No ethical approval was requested. For calculations of contaminants, when an element concentration was under the LOQ, the middle bound (MB) approach was adopted; therefore, its value was assumed to be half of its LOQ. Moreover, in order to perform a quantitative estimation of the severity of potential adverse health effects in the given population, the results of the exposure assessment were compared to the health-based guidance values and expressed as contribution to the Acceptable/Tolerable Daily Intake (ADI/TDI). Furthermore, aiming to quantitatively estimate the health benefits of lake Trasimeno fish consumption, the omega-3 fatty-acid content of these fishes was determined analytically, and the daily dietary intake of such nutrients in the target population was assessed as mentioned above for contaminants. Subsequently, the contribution of the exposure values to the attainment of the suggested Dietary Reference Intake (RDI) of 250 mg/d for EPA and DHA [23] was defined.

### 2.5. Risk–Benefit Assessment

In the present study, the benefit of fish consumption refers mainly to the ingestion of EPA and DHA, identified as active factors in cardiovascular diseases prevention, while risk factors were attributed to the ingestion of contaminants which have been proved toxic to humans.

The benefit–risk quotient (BRQ) was employed to evaluate the risk–benefit of the simultaneous ingestion of PUFA and contaminants through freshwater fish consumption, according to the following equation [24]:(1)$$\mathrm{BRQ}=\frac{Q_{\mathrm{FA}}}{Q_{T}}$$
were Q_FA_ is defined as follows:(2)$$Q_{\mathrm{FA}}=\frac{R_{\mathrm{FA}}}{C_{\mathrm{FA}}}$$
where R_FA_ (mg/d) is the recommended daily intake of EPA + DHA. In this study, the RDI of 250 mg/d for a healthy adult [23] was applied; C_FA_ (mg/g) represents the concentration of EPA + DHA in fish muscles.

The maximum allowable fish consumption related to toxic effects (Q_T_) can be define according to the following equation:(3)$$Q_{T}=\frac{\mathrm{RfD}*\mathrm{bw}}{c}$$
where RfD (mg/kg bw/d) is the reference dose of a pollutant defined through the ADI/TDI of each pollutant considered, bw is the standard bodyweight set, as mentioned above, at 70 kg, and c (mg/g) is the concentration of each toxic molecule in the targeted fish muscle. BRQ values below 1 suggest that achieving the recommended intake of EPA + DHA poses no evident risk to human health linked to the intake of the pollutant through fish consumption [25].

### 2.6. Statistical Analysis

Data were analyzed by descriptive statistics (mean value and standard error of the mean), and an analysis of variance (ANOVA) model was defined using the GLM procedure in SAS version 2001 (SAS institute Inc., Cary, NC, USA). The differences between means were considered to be significant if *p* < 0.05.

## 3. Results and Discussions

The proximate and fatty acid composition and cholesterol content of muscle tissues of freshwater fish and crustaceans collected in this study are presented in Table 1.

The lipid content varied markedly among the fish and crustacean species (0.41%–22.35%). According to the lipid content, the fish species studied were categorized into different groups as lean, low-fat, medium-fat, and high-fat, as reported by other authors [26]. Perch, black bullhead, large-mouth bass, pike, and golden fish are classified in the category of lean fish, as are crawfish (less than 2% fat) lean meat (LM), whereas tench is regarded as a low-fat fish (LFF) (2%–4% fat). Carp and whole sand smelt are classified as medium-fat fish (MFF; 4% > fat < 8%), and eel is classified as high-fat fish (HFF) (over 8% fat) [6,26]. Protein ranged from 14.94% in eel to 18.66% in pike. The ash levels of muscle tissue were similar among species and were approximately 0.99%–1.17%, except for sand smelt. Sand smelt presented an ash amount higher than fillets from the other species because it was analyzed whole. The lipid and protein content present in these fish samples agreed well with the average values reported by other authors for these species [5,10,27]. The moisture content varied from 60.67% in eel to 82.91% in black bullhead. The cholesterol content in this study varied greatly, ranging from a low amount in golden fish to a high amount in sand smelt, equal to 208.4 mg/100 g. However, other fishes presented a high amount of cholesterol, such as eel, with 105.8 mg/100 g, followed by crawfish, with 95.7 mg/100 g, in line with the value reported in the literature. [28]. Cholesterol content in fish also depends on diet, age, sex, spawning cycle, season, and geography; however, a previous study on fish revealed that freshwater fish had a lower cholesterol content than marine fish. [28].

A total of 23 major fatty acids, including eight saturated fatty acids (SFA), six monounsaturated fatty acids (MUFA), and nine PUFA, were identified and quantified. All fatty acids varied significantly among species. Total SFA ranged from 25.29% in the crustacean red swamp crawfish to 36.99% in golden fish, MUFA ranged from 19.55% in perch to 49.67 in eel, and PUFA from 19.59% in eel to 51.26% in pike. The most abundant SFA in all fish studied was palmitic acid (16:0), which varied from 16.2% in crawfish to 22.87% in tench. The second was myristic acid (14:0), which varied from 0.9% in crawfish to 7.7 in sand smelt, and the third was stearic acid (18:0), which varied from 3.6% in sand smelt to 7.5% in golden fish. These values are in agreement with previous studies on freshwater fish from other Italian lakes [2,5,10]. The most represented MUFA in freshwater fish were oleic acid (18:1n-9) (10.3% in perch to 34.6% in eel), 16:1n-7 (2.7% in pike to 11.34% in sand smelt), and 18:1n-7 (2.9% in crawfish to 6.68% in carp). These results are in agreement with the studies on the fatty acid profile of freshwater fish, while a study on marine fish found only palmitoleic (16:1n-7) and oleic (18:1n-9) acids in the main fraction of MUFA [2].

PUFA contents in the studied species have been reported to be in a very wide range. The major fatty acids identified as PUFA were EPA and DHA in all fish species and crawfish. Among PUFA, total n-3 fatty acids ranged from 9.6% in eel to 39.8% in pike. DHA and EPA varied greatly among species. DHA ranged from 3.4% in eel to 29.2% in pike, whereas EPA ranged from 1.7% in eel to 11.6% in crawfish. Perch and pike presented the highest percentages of EPA and DHA, followed by largemouth bass. The lowest percentage of the sum of EPA and DHA was found in eel, followed by black bullhead. The percentages of the EPA and DHA of pike and perch were similar to those reported by other authors [5,29] and comparable to those of some studied marine fish [1,6]. This high percentages of n -3 PUFA, in particular C22:6, have been related by several authors both to the capability of fish to desaturate and elongate enzymatically dietary precursors into long-chain highly unsaturated fatty acids and to their dietary habits [5]. In contrast to fishes, crawfish exhibited a higher EPA level, exceeding that of DHA. These findings are in agreement with those of previous studies on marine crustaceans and are associated with factors characteristic of crustacean physiology [30].

Linoleic acid (18:2n-6) and arachidonic acid (20:4n-6) were the predominant n-6 PUFA in all freshwater fish. Similar results have also been shown in many previous studies of freshwater fish [2]. Linoleic acids varied in fish fillets from 4.32% in pike to 14.9% in crawfish, while arachidonic acid varied from 2.9% in eel to 7.7 in tench; total n-6 fatty acids ranged from 9.98% in eel to 22.6% in crawfish. The ratio n-3/n-6 ranged from 1% in eel to 3.5% in pike. The levels of total n-3 in all the studied freshwater fish species were higher than those of total n-6 PUFAs. The n-3/n-6 ratio has important nutritional significance, and the western diet is characterized by an imbalance of the above-mentioned ratio, with a lower intake of n-3 long-chain PUFA and a concurrent higher intake of n-6 PUFA [31]. An increase in the human dietary n-3/n-6 fatty acid ratio is essential to help prevent cardiovascular disease and reduce cancer risk [32].

To evaluate the effective quantitative level of EPA and DHA introduced by diet, the amounts of EPA and DHA were calculated in fish flesh and expressed as milligrams of fatty acid per 100 g of fish meat (Figure 2).

Eel, sand smelt, carp, and tench, which have a higher fat content than the other fish, contained higher amounts of EPA and DHA (Figure 2). In general, lean fish, such as perch, pike, and large-mouth bass, which have relatively high percentages of the predominant n-3 fatty acids EPA and DHA, showed lower amounts of these fatty acids because of their characteristically low lipid contents. This finding is in agreement with those of some authors [5,33], who found that both marine and freshwater lean fish, such as pollock and hake or perch, respectively, despite containing relatively high percentages of EPA and DHA, are actually poor sources of these fatty acids because of their characteristically low lipid contents.

Table 2 shows the occurrence, middle-bound means, minimum, and maximum concentration values of PCBs, pesticides, and metals in freshwater fish collected from lake Trasimeno.

The PCBs analyzed represent the sum of six congener (PCBs 28, 52, 101, 153, 138, 180) considered as suitable indicators of all PCBs due to their predominance in biotic and non-biotic environments and listed as priority food contaminants monitored by the European Food Safety Authority [34]. Furthermore, linear relationships between indicator PCBs and both dioxin-like PCBs (DL-PCBs) and dioxins have been highlighted in fish [35]. PCBs were detected in 100% of the samples analyzed; nevertheless, all samples showed a low contamination level, well below the European maximum limits for both eel (300 ng/g) and other freshwater fish (75 ng/g) [36].

The residual levels of these chemicals varied among the fish groups considered, ranging from 28 ng/g for the HFF group (represented only by eel) to 1.9 ng/g for LFF. The total PCB concentration in HFF was approximately 15 times higher than that in other fishes, confirming the higher capacity of eel to accumulate PCB congeners. This result is consistent with those reported by other researchers, who found higher levels of PCBs in eel muscle with respect to other species, attributable mainly to the high level of tissue fat peculiar to this species [24]. For this reason, eel is considered a bioindicator of PCB environmental pollution [37].

In contrast to this study, data reported for other countries (Belgium, Germany, France, and the Netherlands) show that the EU limit set for PCB was exceed in eel, which thereby represents a risk to human health. Low levels were found in Polish water bodies; indeed, only 9 of 125 eels exceeded the maximum level. Although PCBs were banned in most developed countries in the 1970s, the extensive and careless use of these compounds for almost 50 years and their persistence have resulted in widespread contamination of the environment and chronic pollution of terrestrial and aquatic ecosystems in Europe [47,48,49].

Regarding pesticides, Table 2 shows only those molecules for which at least one positive sample was found, i.e., pendimethalin and 4,4 DDE. The other investigated molecules were below the limit of detection (LOD) in all fish samples considered (Appendix A). Among all samples analyzed, four LM and one MFF tested positive for pendimethalin, while one LM and two MFF tested positive for 4,4-DDE. The compound 4,4-DDE is the main metabolite of DDT and is considered an indicator of past DDT exposure because 4,4-DDE tends to persist much longer in the environment than DDT. Indeed, Varol and Sünbül [49] reported that 4,4-DDE was the only DDT detected in freshwater fish in Turkey; moreover, both DDT and 4,4-DDE are among the persistent organic pollutants that may cause potential adverse effects on both human and animal health [50]. Their concentrations in fish muscle ranged from 0.002 mg/kg to 0.01 mg/kg. Furthermore, 4,4-DDE was found in goldfish muscle, among other fish species, and the fish muscle concentrations of 4,4-DDE found in this study were lower than those in fish collected from freshwater elsewhere [49,51]. The pesticide pendimethalin is an active herbicide frequently used in terrestrial systems. Due to the common usage of various formulations containing pendimethalin, this chemical compound has been detected at high concentrations in different European aquatic ecosystems [52]. Taken up by fish due to its high bioconcentration factor, pendimethalin exert its toxicity in the human liver [45]. A very low concentration, ranging from 0.002 to 0.004, was found in carp, probably due to the feeding habits of this species.

Table 2 summarizes the trace element concentrations found in the muscles of the fish species from Trasimeno analyzed in the present study. As shown, metal pollutants were found in all four fish groups tested, However, for all the samples considered, the metal content was well below the maximum limits set by the European Commission (1881/2006) [53]. The results suggest that in the sampling season considered in this study, metals concentrated in fish tissues irrespective of species or lipid content. This is in contrast to reports by other authors who highlighted a relation between metal concentration and fish type [24]. The overall mean concentrations of trace metals are considered low; indeed, the highest value was attributable to Hg in HFF (0.227 mg/kg). As already reported in the literature, these results confirm that Hg has a great potential for bioaccumulation in food chains of aquatic ecosystems and, consequently, in edible fish tissues [54,55]. In general, the concentrations of the tested elements were lower than the ranges recorded in other European water environments, such as lakes of Plumbuita (Romania) and Šalek (Slovenia) and lakes of Warmia and Mazury (Poland) [9,54,56].

According to the results of the questionnaire distributed to the target population, freshwater fish accounted for a modest proportion of the consumers’ diets (33 g/d); however, as reported in the literature, it may represent one of the main sources of human exposure to EPA and DHA and, eventually, environmental contaminants [2,24]. The recommended daily intake of EPA + DHA of 250 mg for a healthy adult was achieved with the consumption declared by the consumers of HFF and MFF, which contributed to the RDI for 112% and 109%, while LFF contributed for 62%, and LF for 22%. These results confirm that EPA and DHA concentrations are mainly attributable to the lipid contents of the fish, as reported by other authors.

In order to assess the potential risks associated with the consumption of fish from lake Trasimeno, the mean EDI and the contribution to ADI/TDI were calculated, considering a standard Italian adult weight of 70 kg and the above-mentioned consumption values. The EDIs for PCBs were 1.1, 0.9, 1.4, and 13.2 ng/kg bw/day for LM, LFF, MFF, and HFF fish groups, reflecting a contribution to ADI/TDI below 10% with the exception of the HFF category, which accounted for 66% of the reference value due to the high lipid content of this species. For the two pesticides molecules considered (pendimethalin and 4,4-DDE), the EDI was negligible in all fish groups examined, as was their contributions to the reference value were, in all cases, below 0.000 mg/kg bw/day and below 1%, respectively (Table 2). Similarly, the EDI of the heavy metals assessed was below 0.000 mg/kg bw/day, contributing to the reference value to a very low degree (less than 1%), with the exception of mercury (Hg), with values ranging from 4.3% to 11.3%.

### Risk–Benefit Assessment

Recently, monitoring programs have been implemented by several countries aiming to assess the presence of chemical pollutants in foods as well as to define human health risks resulting from dietary exposure to these contaminants. In the present study, the toxic effects of the targeted contaminants in muscle tissues of lake Trasimeno freshwater fish were evaluated using BRQ. The BRQ for all of the groups of fish analyzed were <1, ranging between 0.00 and 0.59 (Table 2). This strongly suggests that a healthy population, consuming enough fish belonging to any of the four fish groups to achieve the RDI for EPA and DHA, would not be exposed to an increased health risk due to the exposure to the toxicants considered [26]. As already reported by other authors in different environments, these results confirm that the benefits of freshwater fish intake should outweigh the associated risks, considering the average healthy population [7,57].

## 4. Conclusions

The contents of n-3 long-chain PUFAs (EPA and DHA) were higher in species such as eel, sand smelt, carp, and tench and were associated with the lipid contents of the fish, strongly contributing to the RDI of EPA and DHA. In lean fish that showed lower amounts of these fatty acids due to their low lipid contents, the contribution of EPA and DHA to the RDI was more limited than in fatty fish. The concentrations of all of the contaminants that were analyzed in fish muscle tissues were very low, below the limits when present (PCBs, metals), demonstrating the safety of the environmental conditions of the lake. Together, the contribution to reference values and BRQs indicated that the health benefits of consumption of fish from lake Trasimeno outweigh the potential risks; therefore, these species can be consumed regularly by the general population, without posing significant health risks related to the presence of contaminants. The population should be encouraged to consume higher amounts of freshwater fish from lake Trasimeno, as they can be considered a healthy choice and can allow consumers to meet the recommended levels of EPA and DHA.

## Figures and Tables

**Figure 1 ijerph-17-02545-f001:**
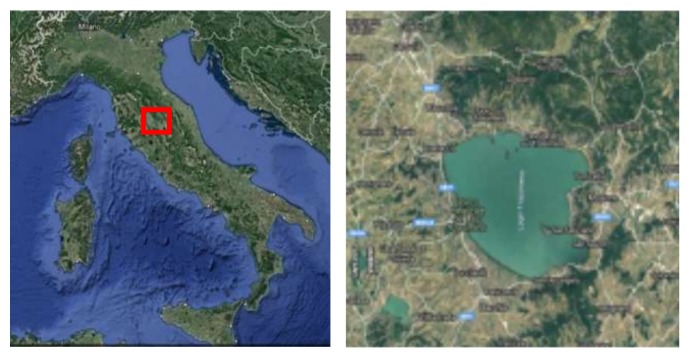
Lake Trasimeno in central Italy (43°08′ N 12°06′ E).

**Figure 2 ijerph-17-02545-f002:**
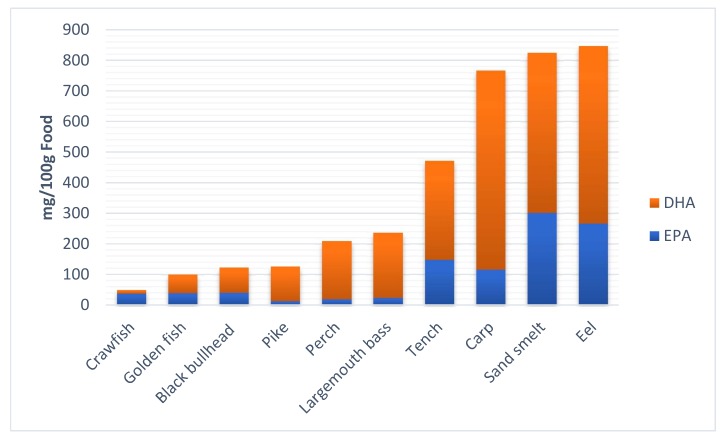
EPA and DHA (mg/100 g food) concentrations in fish meat.

**Table 1 ijerph-17-02545-t001:** Proximate composition (% wet weight), cholesterol (mg/100 g), and fatty acid composition (g/100 g of total fatty acid methyl esters) of fishes of lake Trasimeno (mean ± SD).

	HFF	MFF		LFF	LM					
Species	Eel	Sand Smelt	Carp	Tench	Black Bullhead	Golden Fish	Largemouth Bass	Pike	Perch	Crawfish
*n*	12	18	12	12	12	12	12	12	12	18
Length (cm)	57 ± 12.04	7.08 ± 1.15	78 ± 12.24	42.12 ± 3.45	13 ± 3.05	24.01 ± 1.85	26 ± 2.14	85 ± 15.15	20.04 ± 1.42	8 ± 2.04
Weight (g)	250 ± 22	9 ± 1.14	4220 ± 155	805 ± 55.10	200 ± 36.04	454 ± 35.55	180 ± 20.20	2500 ± 275.5	150 ± 18.14	15 ± 6.12
Ash	1.04 ± 0.06 ^b^	2.35 ± 0.32 ^a^	0.99 ± 0.12 ^b^	1.11 ± 0.07 ^b^	1.01 ± 0.06 ^b^	1.17 ± 0.08 ^b^	1.10 ± 0.04 ^b^	1.11 ± 0.07 ^b^	0.99 ± 0.02 ^b^	1.00 ± 0.08 ^b^
Protein	14.94 ± 0.85 ^a^	16.35 ± 0.59 ^a^	16.23 ± 2.21 ^a^	16.56 ± 1.00 ^ab^	15.15 ± 0.71 ^a^	16.03 ± 0.58 ^a^	18.58 ± 0.68 ^b^	18.66 ± 0.31 ^c^	18.42 ± 1.01 ^b^	15.93 ± 0.48 ^a^
Moisture	60.67 ± 2.04 ^a^	75.90 ± 1.04 ^b^	78.40 ± 2.13 ^c^	79.27 ± 1.00 ^c^	82.91 ± 0.41 ^e^	81.88 ± 0.73 ^de^	79.48 ± 0.48 ^c^	79.45 ± 0.23 ^c^	79.85 ± 1.16 ^cd^	82.64 ± 0.63 ^e^
Lipid	23.35 ± 2.78 ^d^	5.4 ± 1.46 ^c^	4.38 ± 1.58 ^c^	3.06 ± 1.64 ^bc^	0.93 ± 0.52 ^ab^	0.91 ± 0.34 ^ab^	0.84 ± 0.44 ^ab^	0.78 ± 0.13 ^ab^	0.75 ± 0.25 ^ab^	0.41 ± 0.16 ^a^
Cholesterol	105.73 ± 4.66 ^f^	208.37 ± 5.55 ^g^	53.61 ± 1.64 ^ab^	54.72 ± 2.45 ^b^	64.50 ± 1.80 ^c^	48.48 ± 1.69 ^a^	69.35 ± 2.08 ^c^	58.28 ± 1.24 ^b^	75.99 ± 4.12 ^d^	95.62 ± 3.09 ^e^
**Fatty acids**										
C12:0	0.11 ± 0.01 ^ab^	0.24 ± 0.14 ^ab^	0.10 ± 0.05 ^ab^	0.25 ± 0.16 ^ab^	0.25 ± 0.11 ^ab^	0.45 ± 0.35 ^b^	0.10 ± 0.06 ^ab^	0.08 ± 0.05 ^a^	0.35 ± 0.24 ^ab^	0.25 ± 0.13 ^ab^
C14:0	4.78 ± 0.35 ^d^	7.70 ± 1.33 ^e^	2.59 ± 0.58 ^abc^	3.46 ± 1.17 ^bcd^	2.83 ± 0.96 ^abcd^	3.73 ± 1.76 ^cd^	2.75 ± 1.21 ^abc^	1.54 ± 0.22 ^ab^	1.92 ± 0.67 ^abc^	0.90 ± 0.46 ^a^
C15:0	0.55 ± 0.04	0.95 ± 0.13	1.07 ± 0.13	0.55 ± 0.32	0.65 ± 0.41	1.02 ± 0.39	0.48 ± 0.26	0.55 ± 0.31	0.70 ± 0.24	0.75 ± 0.15
C16:0	20.67 ± 1.18 ^ab^	17.74 ± 1.30 ^a^	17.80 ± 3.42 ^a^	22.87 ± 4.20 ^b^	17.34 ± 1.33 ^a^	22.85 ± 3.22 ^b^	21.38 ± 2.00 ^ab^	17.26 ± 2.07 ^a^	20.45 ± 0.91 ^ab^	16.17 ± 1.15 ^a^
C17:0	0.37 ± 0.15 ^a^	0.86 ± 0.22 ^ab^	1.45 ± 0.59 ^b^	0.89 ± 0.22 ^ab^	0.73 ± 0.45 ^a^	0.92 ± 0.24 ^ab^	0.67 ± 0.30 ^a^	0.67 ± 0.29 ^a^	0.75 ± 0.28 ^a^	0.85 ± 0.12 ^ab^
C18:0	4.07 ± 0.13 ^a^	3.58 ± 0.49 ^a^	4.91 ± 1.94 ^ab^	5.71 ± 0.79 ^ab^	5.13 ± 2.02 ^ab^	7.46 ± 1.12 ^b^	5.53 ± 1.68 ^ab^	6.88 ± 2.46 ^ab^	6.34 ± 1.54 ^b^	6.00 ± 1.42 ^ab^
C20:0	0.09 ± 0.07	0.13 ± 0.07	0.21 ± 0.14	0.11 ± 0.07	0.19 ± 0.15	0.36 ± 0.30	0.26 ± 0.19	0.05 ± 0.04	0.19 ± 0.16	0.22 ± 0.10
C21:0	0.11 ± 0.07 ^a^	0.21 ± 0.16 ^ab^	0.07 ± 0.05 ^a^	0.13 ± 0.08 ^ab^	0.20 ± 0.18 ^ab^	0.21 ± 0.19 ^ab^	0.49 ± 0.35 ^b^	0.31 ± 0.25 ^ab^	0.22 ± 0.12 ^ab^	0.15 ± 0.09 ^ab^
**Total SFA**	30.74 ± 1.64 ^abc^	31.40 ± 1.99 ^abc^	28.19 ± 4.64 ^ab^	33.98 ± 5.66 ^bc^	27.31 ± 3.54 ^ab^	36.99 ± 4.94 ^c^	31.66 ± 1.77 ^abc^	27.34 ± 3.78 ^ab^	30.93 ± 2.49 ^abc^	25.29 ± 2.15 ^a^
C16:1	10.06 ± 0.53 ^c^	11.34 ± 1.61 ^c^	9.93 ± 2.98 ^c^	7.48 ± 2.87 ^b^	7.48 ± 1.48 ^b^	5.97 ± 1.54 ^ab^	6.04 ± 2.26 ^ab^	2.74 ± 0.84 ^a^	4.61 ± 0.99 ^ab^	3.61 ± 0.24 ^a^
C18:1n9cis	34.59 ± 0.52 ^e^	15.30 ± 2.05 ^bc^	13.72 ± 1.51 ^abc^	15.75 ± 3.96 ^bc^	18.08 ± 2.53 ^dc^	16.58 ± 2.98 ^bc^	12.98 ± 1.59 ^ab^	13.69 ± 1.71 ^abc^	10.34 ± 2.02 ^a^	21.89 ± 1.60 ^d^
C18:1n7cis	4.13 ± 0.27 ^ab^	4.39 ± 0.44 ^ab^	6.68 ± 0.85 ^d^	4.12 ± 1.16 ^ab^	6.02 ± 0.93 ^cd^	4.62 ± 1.00 ^bc^	4.31 ± 0.56 ^ab^	4.06 ± 0.99 ^ab^	3.64 ± 0.39 ^ab^	2.86 ± 0.14 ^a^
C20:1	0.79 ± 0.32 ^ab^	0.43 ± 0.57 ^ab^	1.18 ± 0.84 ^b^	0.92 ± 0.51 ^ab^	0.63 ± 0.39 ^ab^	0.35 ± 0.25 ^ab^	0.33 ± 0.18 ^b^	0.58 ± 0.45 ^ab^	0.23 ± 0.13 ^a^	0.24 ± 0.11 ^a^
C22:1n9	0.04 ± 0.02 ^a^	0.07 ± 0.05 ^ab^	0.36 ± 0.29 ^b^	0.17 ± 0.12 ^ab^	0.13 ± 0.11 ^ab^	n.d.	0.22 ± 0.19 ^ab^	0.04 ± 0.02 ^a^	0.33 ± 0.16 ^b^	n.d.
C24:1	0.07 ± 0.03 ^a^	0.19 ± 0.18 ^ab^	0.17 ± 0.16 ^ab^	0.19 ± 0.17 ^ab^	0.16 ± 0.14 ^ab^	0.27 ± 0.23 ^ab^	0.38 ± 0.22 ^ab^	0.29 ± 0.28 ^ab^	0.40 ± 0.15 ^ab^	0.58 ± 0.35 ^b^
**Total MUFA**	49.67 ± 0.86 ^d^	31.72 ± 1.23 ^c^	32.03 ± 2.75 ^c^	28.64 ± 6.76 ^bc^	32.50 ± 3.57 ^c^	27.78 ± 4.23 ^bc^	24.26 ± 3.16 ^ab^	21.39 ± 1.32 ^ab^	19.55 ± 1.74 ^a^	29.17 ± 2.00 ^bc^
C18:2n6*cis*	6.46 ± 0.23 ^b^	7.74 ± 1.10 ^b^	4.58 ± 0.44 ^ab^	8.70 ± 1.60 ^bc^	12.68 ± 4.23 ^c^	8.69 ± 2.48 ^bc^	8.41 ± 3.11 ^b^	4.32 ± 0.52 ^ab^	4.13 ± 1.17 ^ab^	14.98 ± 2.34 ^c^
C18:3n6	0.16 ± 0.02 ^ab^	0.62 ± 0.12 ^c^	0.24 ± 0.16 ^ab^	0.18 ± 0.12 ^ab^	0.20 ± 0.14 ^ab^	0.23 ± 0.11 ^ab^	0.37 ± 0.12 ^b^	0.12 ± 0.07 ^a^	0.17 ± 0.12 ^a^	0.23 ± 0.12 ^ab^
C18:3n3	2.55 ± 0.11 ^ab^	5.79 ± 0.74 ^cd^	3.03 ± 0.26 ^ab^	2.28 ± 1.22 ^ab^	7.83 ± 3.44 ^d^	4.24 ± 0.91 ^bc^	2.51 ± 1.24 ^ab^	2.14 ± 0.33 ^ab^	1.30 ± 0.25 ^a^	4.25 ± 1.03 ^bc^
C20:3n6	0.48 ± 0.08 ^abc^	0.30 ± 0.12 ^ab^	0.77 ± 0.13 ^c^	0.56 ± 0.39 ^bc^	0.73 ± 0.27 ^c^	0.34 ± 0.33 ^abc^	0.41 ± 0.08 ^abc^	0.10 ± 0.09 ^a^	0.30 ± 0.23 ^ab^	0.32 ± 0.11 ^abc^
C20:4n6	2.88 ± 0.12 ^a^	4.11 ± 0.22 ^ab^	6.16 ± 0.98 ^bc^	7.71 ± 3.08 ^bc^	4.50 ± 1.47 ^ab^	5.51 ± 0.92 ^abc^	6.60 ± 1.58 ^bc^	6.98 ± 1.24 ^bc^	7.98 ± 0.57 ^bc^	7.06 ± 1.54 ^bc^
C20:3n3	0.35 ± 0.16 ^a^	0.33 ± 0.02 ^a^	0.54 ± 0.20 ^ab^	0.23 ± 0.16 ^a^	0.75 ± 0.30 ^b^	0.29 ± 0.26 ^a^	0.28 ± 0.16 ^a^	0.22 ± 0.15 ^a^	0.30 ± 0.18 ^a^	0.28 ± 0.17 ^a^
C20:5n3 (EPA)	1.65 ± 0.12 ^a^	5.11 ± 0.61 ^b^	3.55 ± 0.83 ^ab^	4.68 ± 2.86 ^b^	4.29 ± 1.58 ^ab^	4.64 ± 0.95 ^b^	2.32 ± 0.90 ^ab^	4.55 ± 0.81 ^ab^	4.51 ± 1.69 ^ab^	11.59 ± 2.16 ^c^
C22:5n3	1.69 ± 0.24 ^ab^	3.72 ± 0.23 ^c^	2.90 ± 0.94 ^bc^	2.75 ± 1.15 ^bc^	2.45 ± 0.82 ^bc^	2.11 ± 0.84 ^ab^	3.61 ± 0.59 ^bc^	3.70 ± 0.14 ^c^	3.69 ± 1.18 ^c^	0.77 ± 0.47 ^a^
C22:6n3 (DHA)	3.38 ± 0.44 ^a^	9.15 ± 2.00 ^b^	18.02 ± 1.68 ^c^	10.31 ± 4.50 ^b^	6.75 ± 1.74 ^ab^	9.17 ± 2.05 ^b^	19.57 ± 4.24 ^c^	29.15 ± 1.59 ^d^	27.15 ± 2.34 ^d^	6.07 ± 1.06 ^ab^
**Total PUFA**	19.59 ± 1.08 ^a^	36.88 ± 2.53 ^bc^	39.78 ± 4.05 ^bc^	37.39 ± 10.76 ^bc^	40.19 ± 6.04 ^bc^	35.23 ± 4.32 ^bc^	44.08 ± 3.12 ^cd^	51.26 ± 3.81 ^d^	49.52 ± 1.22 ^cd^	45.54 ± 3.81 ^cd^
n-3 PUFA	9.62 ± 0.89 ^a^	24.11 ± 2.33 ^bc^	28.04 ± 3.08 ^c^	20.24 ± 7.64 ^b^	22.08 ± 3.42 ^bc^	20.46 ± 2.66 ^b^	28.28 ± 4.33 ^c^	39.76 ± 1.98 ^d^	36.94 ± 1.91 ^d^	22.95 ± 2.17 ^bc^
n-6	9.98 ± 0.25 ^a^	12.77 ± 1.07 ^abc^	11.74 ± 1.18 ^ab^	17.15 ± 3.47 ^c^	18.11 ± 3.03 ^cd^	14.77 ± 2.88 ^bc^	15.79 ± 3.09 ^bc^	11.51 ± 1.84 ^ab^	12.58 ± 0.89 ^ab^	22.59 ± 1.86 ^d^
n-3/n-6	0.96 ± 0.08 ^a^	1.90 ± 0.23 ^bc^	2.39 ± 0.18 ^c^	1.16 ± 0.24 ^a^	1.23 ± 0.15 a	1.42 ± 0.25 ^ab^	1.87 ± 0.56 ^bc^	3.52 ± 0.50 ^d^	2.96 ± 0.36 ^cd^	1.02 ± 0.06 ^a^

Categorization of fish based on lipid content: lean fish, (lean meat, LM) fat <2%; low-fat fish (LFF) (fat = 2%–4%); medium-fat fish (fat = 4% > MFF < 8%); high-fat fish (HFF) (fat > 8%); n, number of specimens analyzed. Results are the mean values ± standard deviation; values within a row with different letters (^a b c d e f g^) are significantly different (*p* < 0.05). SFA, saturated fatty acids; MUFA, monounsaturated fatty acids; PUFA, polyunsaturated fatty acids; EPA, eicosapentaenoic acid; DHA, docosahexaenoic acid; n.d., not detected.

**Table 2 ijerph-17-02545-t002:** Incidence of contaminants in fish, average contamination as middle bound (MB, mg/kg), estimated daily intake (EDI g/kg bw/day), contribution to the health-based guidance values Acceptable/Tolerable Daily Intake (ADI/TDI), and benefit–risk quotient (BRQ).

	Ndetected/Total	Min	Max	Average (MB)	EDI	% ADI/TDI **	BRQ
*Pendimethalin*							
LM	4/16	<0.002	0.010	0.002	0.000	0.001	0.00
LFF	0/9	<0.002	-	0.001	0.000	0.000	0.00
MFF	1/3	<0.002	0.006	0.003	0.000	0.001	0.00
HFF	0/3	<0.002	-	0.001	0.000	0.000	0.00
*4.4-DDE*							
LM	1/18	<0.002	0.002	0.002	0.000	0.009	0.00
LFF	0/10	<0.002	-	0.001	0.000	0.005	0.00
MFF	2/4	<0.002	0.004	0.002	0.000	0.009	0.00
HFF	0/3	<0.002	-	0.001	0.000	0.005	0.00
*Pb*							
LM	9/22	<0.015	0.197	0.020	0.000	0.264	0.01
LFF	0/0	<0.015	-	0.007	0.000	0.092	0.00
MFF	8/30	<0.015	0.068	0.013	0.000	0.172	0.00
HFF	1/10	<0.015	0.018	0.008	0.000	0.106	0.00
*Hg*							
LM	22/22	0.048	0.206	0.123	0.000	10.148	0.46
LFF	6/6	0.033	0.065	0.052	0.000	4.290	0.07
MFF	27/30	<0.025	0.119	0.067	0.000	5.528	0.05
HFF	10/10	0.049	0.227	0.137	0.000	11.303	0.10
*Cd*							
LM	18/22	<0.005	0.013	0.004	0.000	0.538	0.01
LFF	1/6	<0.005	0.008	0.003	0.000	0.502	0.01
MFF	0/25	<0.005	-	0.002	0.000	0.467	0.01
HFF	1/5	<0.005	0.022	0.002	0.000	0.467	0.01
*Ni*							
LM	0/22	<0.040	-	0.020	0.000	0.337	0.02
LFF	0/1	<0.040	-	0.020	0.000	0.337	0.02
MFF	14/22	<0.040	0.153	0.041	0.000	0.690	0.01
HFF	0/3	<0.040	-	0.020	0.000	0.337	0.02
*Cr*							
LM	2/3	<0.040	0.040	0.020	0.000	0.003	0.14
LFF	1 /1	-	0.040	0.020	0.000	0.003	0.14
MFF	0/20	<0.040	-	0.020	0.000	0.003	0.14
HFF	0/5	<0.040	-	0.020	0.000	0.003	0.14
*As*							
LM	4/4	0.105	0.152	0.136	0.000	0.049	0.00
LFF	1/1	-	0.056	0.056	0.000	0.020	0.00
MFF	19/20	<0.040	0.168	0.077	0.000	0.028	0.00
HFF	2/5	<0.040	0.075	0.038	0.000	0.016	0.000
*PCB* *							
LM	12/12	0.890	9.390	2.360	1.113	5.563	0.25
LFF	5/5	1.400	2.580	1.960	0.924	4.620	0.07
MFF	3/3	2.020	4.370	2.900	1.367	6.836	0.07
HFF	11/11	3.550	84.110	28.000	13.200	66.000	0.59

* Concentration values for polychlorinated biphenyls (PCBs) are expressed as ng/g of wet tissue, EDI is expressed as ng/kg bw/d. ** health-based guidance values: Pb = 0.004 mg/kg bw/d (FAO/WHO, 2001) [38], Hg = 0.571 ug/kg bw/d (EFSA, 2018) [39], Cd = 0.35 ug/kg bw/d (EFSA, 2011) [40] , Ni = 2.8 ug/kg bw/d (EFSA, 2015) [41] , Cr = 0.3 mg/kg bw/d (EFSA, 2014) [42], As = 130 ug/kg bw/d (Parviz et al., 2015) [43], PCBs = 20 ng/kg bw/d (Arnich et al., 2009) [44], pendimenthalin = 0.125 mg/kg bw/d (EFSA, 2016) [45], and ppDDE = 10 ug/kg bw/d (FAO/WHO, 2000) [46].

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
