# Peer review of "Nutritional Value and Contaminant Risk Assessment of Some Commercially Important Fishes and Crawfish of Lake Trasimeno, Italy"

_ijerph, 2020, doi:10.3390/ijerph17072545_

Round 1

Reviewer 1 Report

The paper is interesting and the scientific quality of the presentation is high. The applied analytical methods are well explained. In my view, the author should discuss more (even in the abstract) if the consumption of  the selected specimens and the small-scale traditional fishing in this "Site of  Community Interest" may be considered sustainable and maybe  a "model" for a wider audience, highlighting pro&cons. Please add in Table 1 the number of specimens analyzed in each column (it may be deduced from the experimental section, but is not totally clear: how many carps were analyzed? 12? 18?)

Author Response

Reviewer 1 comments

Yes

Can be improved

Must be improved

Not applicable

Does the introduction provide sufficient background and include all relevant references?

(x)

( )

( )

( )

Is the research design appropriate?

(x)

( )

( )

( )

Are the methods adequately described?

(x)

( )

( )

( )

Are the results clearly presented?

(x)

( )

( )

( )

Are the conclusions supported by the results?

(x)

( )

( )

( )

Reviewer 1 comments and suggestions

The paper is interesting and the scientific quality of the presentation is high. The applied analytical methods are well explained. In my view, the author should discuss more (even in the abstract) if the consumption of the selected specimens and the small-scale traditional fishing in this "Site of  Community Interest" may be considered sustainable and maybe  a "model" for a wider audience, highlighting pro&cons.

Author response to reviewer 1 comments and suggestions

The authors thank the reviewer for the interesting suggestion and an aspect of it was added in the manuscript.

The small-scale traditional fishing, as well as the consumption of fish in the area of lake Trasimeno, can be considered a sustainable. In fact the freshwater fish industry of this area is expanding into the market by exploiting the great potential of abundant undervalued species consumed as ingredients in processed food and by the adoption of long term preservation through a cold chain system to compensate the seasonality of fish. Lines 96-101.

On the other hand the “modelization” on a wider scale of this study is hardly applicable due to the peculiar characteristics of the lake Trasimeno. Specifically, the factors determining this scenario are attributable to:  the relatively low level of contaminants due to Special Protection Zone and Regional Park attribution of Trasimeno lake area; the low level of antropization of the lake area which strongly influences the characteristics of the lacustrine ecosystem; the balance among aquatic species that is influenced by specific repopulating plans. Furthermore, Trasimeno is a laminar lake differently from the majority of Italian lakes. In the authors opinion, in the light of the above mentioned consideration, the Trasimeno Lake represents a confined case-study not suitable for modelization or wider applications hard to be discussed in the manuscript. If the reviewer does not consider it needful the authors would not to discuss this aspect in the manuscript.  Furthermore due to the guideline of the journal concerning the abstract length the author find difficult to insert further discussion in this section of the manuscript

Reviewer 1 comments and suggestions

Please add in Table 1 the number of specimens analyzed in each column (it may be deduced from the experimental section, but is not totally clear: how many carps were analyzed? 12? 18?)

Author response to reviewer 1 comments and suggestions

The correction was made as suggested by the reviewer, the number of specimens analyzed were added in each column of table 1.

Reviewer 2 Report

In the paper “Nutritional value and contaminant risk assessment of commercially important freshwater fish species of Trasimeno Lake, Italy” the Authors reported a benefit-risk assessment associated to the consumption of crawfish and nine fish species from the Italian Lake Trasimeno. They carried out analysis of both muscle fatty acids on one hand, and heavy metals and persistent organic pollutants on the other hand.   

In this work results important from a sanitary viewpoint, aimed at improving both food habits of local population and local economy, were discussed. The paper is generally well-written and easy to read and understand.

Major comments:

- I did not understand what pesticides you analysed. You mentioned organochlorine pesticides (OCPs) throughout the manuscript. In the methods section you mentioned also organophosphorus and pyrethroids. You considered Pendimethalin an OCP but it isn’t. I suggest you to include the whole list of pesticides you analysed as supplementary material and to use the term “pesticides” instead of “OCPs” in the text.

- For the three categories of pollutants analysed (polychlorinated biphenyls, heavy metals, pesticides), you did not report the same detail of information for quality control and quality assurance (% recovery of internal standards, LOQ values,…).

 - Pay attention to the use of abbreviations or acronyms throughout the text. If you use them you should write the whole definition just the first time and then use abbreviations or acronyms.

Minor comments:

- The crawfish species should be included in the title and also throughout the manuscript. You cannot refer to it as a freshwater fish. E.g. “Nutritional value and contaminant risk assessment of commercially important fishes and crawfish from Lake Trasimeno, Italy”

- Lines 17-21: It is a too long sentence (pay attention to this throughout the manuscript). It may be modified as follows: “The aim of our study was to describe the balance between health benefits and risks due to the consumption of crawfish and nine fish species from Lake Trasimeno. We thus determined both muscle fatty acid profiles (particularly eicosapentaenoic and docosahexaenoic acids) and chemical pollutants (some polychlorinated biphenyls, pesticides and heavy metals).”

- Lines 28 and 381: Remove “aquatic”

- Line 30: “Lake Trasimeno”, correct it throughout the text.

- Lines 44, 45, 57: “food web”

- Line 46: “fishes”, check it throughout the text.

- Lines 59-60: “each country considers… assesses…”

- Line 66: “estimated daily intake (EDI)”

- Line 70: “(Figure 1)”

- Figure 1: You may include a map of the whole Italy with a symbol showing the lake position instead of the photo on the left.

- Line 100: “Fishes and crawfish sampling was…”

- Line 103: Remove “fish”. “each individual was…”

- Lines 107-108: Remove “collected in… the analysis”

- Line 109: “of each individual were determined according”

- Line 117: “weighted”

- Line 118: “and freeze-dried”

- Line 160: “The EDI of metals…”

- The format of Table 1 needs to be improved, also the footnotes should be separated from the main text.

- Line 217: Remove the repetition “according to the fat content”

- Line 229: Add the reference.

- Line 257: “exhibited”

- Line 262: “arachidonic acid”

- Line 272: “(Figure 2)”

- Table 2 caption: “contaminants … quotient (BRQ)”, also specify what is MB (middle bound?) and the measurement unit of concentrations (mg/g).

- Table 2 footnotes: Do the two asterisks refer to EDI?

- Line 299: “Sigma?”

- Line 301: Add reference, “(Commission Regulation EC N° 1259/2011)”?

- Line 315: “Regarding pesticides, Table 2 shows…”

- Line 329: “ecosystems”

- Line 330: “exerts its”

- Line 345: “accounted”

- Line 348: “of HFF and…”

- Line 349: “contributed”

- Line 350: “are mainly”

- Line 353: “were calculated”

- Line 369: “suggests”

Author Response

Reviewer 2 comments

Yes

Can be improved

Must be improved

Not applicable

Does the introduction provide sufficient background and include all relevant references?

(x)

( )

( )

( )

Is the research design appropriate?

(x)

( )

( )

( )

Are the methods adequately described?

( )

(x)

( )

( )

Are the results clearly presented?

(x)

( )

( )

( )

Are the conclusions supported by the results?

(x)

( )

( )

( )

Author response to reviewer 2 comments

The authors improve the description of the  methods adding  informations for quality control and quality assurance  for PCB and metals see point 2

Reviewer 2 comments and suggestions and Response to referee’s comments and suggestions

In the paper “Nutritional value and contaminant risk assessment of commercially important freshwater fish species of Trasimeno Lake, Italy” the Authors reported a benefit-risk assessment associated to the consumption of crawfish and nine fish species from the Italian Lake Trasimeno. They carried out analysis of both muscle fatty acids on one hand, and heavy metals and persistent organic pollutants on the other hand.   

In this work results important from a sanitary viewpoint, aimed at improving both food habits of local population and local economy, were discussed. The paper is generally well-written and easy to read and understand.

No comment required

Major comments:

1)- I did not understand what pesticides you analysed. You mentioned organochlorine pesticides (OCPs) throughout the manuscript. In the methods section you mentioned also organophosphorus and pyrethroids. You considered Pendimethalin an OCP but it isn’t. I suggest you to include the whole list of pesticides you analysed as supplementary material and to use the term “pesticides” instead of “OCPs” in the text.

1)The authors want to thanks the reviewer for the suggestion, the term OCPs was changed with pesticides along the text

The authors as suggested by the reviewer include the whole list of pesticide analysed as supplementary material Table S1

2) For the three categories of pollutants analysed (polychlorinated biphenyls, heavy metals, pesticides), you did not report the same detail of information for quality control and quality assurance (% recovery of internal standards, LOQ values,…).

2)The authors, as suggested by the reviewer, added the information for quality control in the text (Lines 135-140; Lines 146-148)

3) Pay attention to the use of abbreviations or acronyms throughout the text. If you use them you should write the whole definition just the first time and then use abbreviations or acronyms.

3)The abbreviations or acronyms were all checked throughout the text

Minor comments:

4) The crawfish species should be included in the title and also throughout the manuscript. You cannot refer to it as a freshwater fish. E.g. “Nutritional value and contaminant risk assessment of commercially important fishes and crawfish from Lake Trasimeno, Italy”

4)The correction was done as suggested

5)Lines 17-21: It is a too long sentence (pay attention to this throughout the manuscript). It may be modified as follows: “The aim of our study was to describe the balance between health benefits and risks due to the consumption of crawfish and nine fish species from Lake Trasimeno. We thus determined both muscle fatty acid profiles (particularly eicosapentaenoic and docosahexaenoic acids) and chemical pollutants (some polychlorinated biphenyls, pesticides and heavy metals).”

5)The correction was done as suggested, and the sentence was rephrased accordingly (Lines 17-20).

6) Lines 28 and 381: Remove “aquatic”

6)The correction was done as suggested.

7) Line 30: “Lake Trasimeno”, correct it throughout the text.

7)The correction was done as suggested

8) Lines 44, 45, 57: “food web”     

8)The correction was done as suggested (Lines 43, 55)

9 Line 46: “fishes”, check it throughout the text.

9)The correction was done as suggested and throughout the text

10) Lines 59-60: “each country considers… assesses…”

10)The correction was done as suggested. Line 58

11)Line 66: “estimated daily intake (EDI)”

11)The correction was done as suggested. Line 65

12) Line 70: “(Figure 1)”

12)The correction was done as suggested. Line 72

13) Figure 1: You may include a map of the whole Italy with a symbol showing the lake position instead of the photo on the left  

13)The correction was made as suggested and a map of the whole Italy with a symbol showing the lake position was added.

14) Line 100: “Fishes and crawfish sampling was…”

14)The correction was done as suggested (Lines107-108).

15) Line 103: Remove “fish”. “each individual was…”

15)The correction was done as suggested. Line 110

16) Lines 107-108: Remove “collected in… the analysis”

16)The correction was done as suggested. Lines110-111

17) Line 109: “of each individual were determined according”

17)The correction was done as suggested. Lines 115

18) Line 117: “weighted”

18)The correction was done as suggested. Line 123

19) Line 118: “and freeze-dried”

19)The correction was done as suggested. Line 124

20) Line 160: “The EDI of metals…”

20)The correction was done as suggested. Line 174

21) The format of Table 1 needs to be improved, also the footnotes should be separated from the main text.

21)The correction was done as suggested

22) Line 217: Remove the repetition “according to the fat content”

22)The correction was done as suggested. Line 244

23) Line 229: Add the reference.

23)The correction was done as suggested. Line 256

24) Line 257: “exhibited”

24)The correction was done as suggested. Line 282

25) Line 262: “arachidonic acid”

25) correction was done as suggested. Line 287

26)Line 272: “(Figure 2)”

26)The correction was done as suggested. Line 297

27)Table 2 caption: “contaminants … quotient (BRQ)”, also specify what is MB (middle bound?) and the measurement unit of concentrations (mg/g).

27)The corrections were done as suggested.

28) Table 2 footnotes: Do the two asterisks refer to EDI?

 28)The authors thank the reviewer for the comment, the two asterisks refer to the ADI/TDI. The asterisks were positioned accordingly

29) Line 299: “Sigma?”

29)The misprint was deleted

30) Line 301: Add reference, “(Commission Regulation EC N° 1259/2011)”

30)The correction was done as suggested  and the reference was added

31) Line 315: “Regarding pesticides, Table 2 shows…”

 31)The correction was done as suggested. Line 344

32) Line 329: “ecosystems”

32)The correction was done as suggested. Line 357

33) Line 330: “exerts its”

33)The correction was done as suggested. Line 358

34) Line 345: “accounted”

34)The correction was done as suggested. Line 375

35) Line 348: “of HFF and…”

 35)The correction was done as suggested. Line 378

36) Line 349: “contributed”

36)The corrections was done as suggested. Line 379

37) Line 350: “are mainly”

37)The correction was done as suggested. Line 380

38) Line 353: “were calculated”

38)The correction was done as suggested. Line 383

39) Line 369: “suggests”

39)The correction was done as suggested. Line 399              

Reviewer 3 Report

Dear authors, 

you can find my report attached.

Best regards

Author Response

Reviewer 3 Comments

Yes

Can be improved

Must be improved

Not applicable

Does the introduction provide sufficient background and include all relevant references?

(x)

( )

( )

( )

Is the research design appropriate?

( )

(x)

( )

( )

Are the methods adequately described?

(x)

( )

( )

( )

Are the results clearly presented?

(x)

( )

( )

( )

Are the conclusions supported by the results?

(x)

( )

( )

( )

Author response to reviewer comment

Please see point 4

Reviewer 3 comments and suggestions and Response to referee’s comments and suggestions

General comments: The paper by Raffaella Branciari and colleagues focuses on a very interesting topic for the health of people living around Trasimeno Lake, ad of course for the literature of the field. The contribute of their paper can represent a good starting point both as reference data for the studied environments and as tools for aquatic food and habitats management. The present study has a good experimental idea and the paper is well written and drafted.

No comment required

1)Title: The title of the paper is well focused, if the authors are sure that the 10 species analyzed are all commercially important species in the area, otherwise i suggest to consider the use of “some commercially important..”.

1)The correction was done as suggested

2)Introduction: In the present manuscript, the authors well depict the knowledge state of art about the focused topics.

2)No comment required

3)Materials and Methods; The choice of the sampling period, although falling within a single season and not over the course of a year, is correct as it is the season following the rains where the contaminants in the lake should be greater. However, it remains a weakness in the sampling plan, that makes the study valid but not absolutely complete.

3)The authors agree with reviewer’s consideration. The choice of the investigation during a specific season can be attributable also to the seasonality of fish; in the selected season all the investigated species were available or allowed to be fished.

4)Results and Discussions; In the discussion section the authors support with good references their results.

4)No comment required

5)Lines 335-337: authors should consider to correlate this sentences to their sampling season. The cited paper to which they refer (Nutrients and contaminants in tissues of five fish species obtained from Shanghai markets: Risk–benefit evaluation from human health perspectives. Geng et al., 2015) result in any case more reliable since it is based on data obtained from two different sampling seasons (winter and summer). In fact, as Geng and colleague reported in introduction: “Considering that the seasonal variations in fish body compositions could affect contaminant concentrations in fish tissues, samples were collected in both winter and summer.”

5)The sentence was correlated to season as suggested. Lines 364-365

6)Line 370: why “This strongly suggest that healthy population, consuming enough fish belonging to any of the five fish groups.” if you consider four groups during all the study? As you reported in lines 216-217 “According to the lipid content, the fish species studied were categorized into different groups as lean, low fat, medium fat and high fat according to the fat content as reported by other authors [24]”.

6)The groups were actually four and the misprint was corrected. Line 399

 7)Conclusions; Conclusions section don’t needs revisions in my opinion

7) No comment required
